# Effect of Levels of Self-Regulation and Situational Stress on Achievement Emotions in Undergraduate Students: Class, Study and Testing

**DOI:** 10.3390/ijerph17124293

**Published:** 2020-06-16

**Authors:** Jesús de la Fuente, Paola Verónica Paoloni, Manuel Mariano Vera-Martínez, Angélica Garzón-Umerenkova

**Affiliations:** 1School of Education and Psychology, University of Navarra, 31009 Pamplona, Spain; 2School of Psychology, University of Almería, 04120 Almería, Spain; 3CONICET (National Scientific and Technical Research Council)—National University of Río Cuarto, Cordoba 5800, Argentina; paopaoloni17@hotmail.com; 4University School La Inmaculada, University of Granada, Calle Joaquina Eguaras, 114, 18013 Granada, Spain; mmvera@ugr.es; 5School of Psychology, Fundación Universitaria Konrad Lorenz, Cra. 9 Bis #62-43, 110231 Bogotá, Colombia; agarzonu@gmail.com

**Keywords:** achievement emotions, self-regulation behavior, academic stress situations, undergraduate students, emotional well-being

## Abstract

Achievement emotions constitute one important variable among the many variables of students’ learning. The aim of this research was to analyze the differential effect of university students’ levels of self-regulation (1 = low, 2 = medium and 3 = high), and of their level of perceived stress in three academic situations (1 = class, 2 = study time and 3 = testing), on the type of achievement emotionality they experience (positive and negative emotions). The following hypotheses were established: (1) a higher level of student self-regulation would be accompanied by higher levels of positive emotionality and lower levels of negative emotionality and (2) a higher level of situational stress would predispose higher levels of negative emotionality and lower levels of positive emotionality. A total of 520 university students completed three self-reports with validated inventories. Descriptive, correlational, and structural prediction analyses (SEM) were performed, as well as 3 × 3 ANOVAs, under an ex post facto design by selection. The results showed overall fulfillment of the hypotheses, except for a few specific emotions. Implications for prevention and psychoeducational guidance in the sphere of university education are discussed.

## 1. Introduction

University, or higher education, seeks to offer adequate scientific and vocational preparation. This objective, however, has been expanded to include elements of emotional well-being. Increasingly, assessment of university quality incorporates satisfaction with the university experience, and how it provides for psychological well-being [1,2]. To better understand this reality, recent research has taken a shift towards analyzing and understanding individual differences, in interaction with different academic situations. The objective is to delimit the type of emotional experiences that such situations provide, and to understand how these experiences act as an indicator of students’ emotional well-being [3]. The present research study, therefore, is situated at the molar level of psychoeducational analysis [4]. This means that the research problem centers on the person × context level of analysis, in a real, ecologically valid situation. The molar level of research focuses on the person, and may exclude the context, while the microanalysis level analyzes discrete, basic processes of behavior.

### 1.1. Emotion Regulation as Individual Variable of Learning in University Students

Students’ self-regulation behavior has been considered a personality-related variable; in other words, it stems from the characteristics of one’s personality. It has a high positive correlation with conscientiousness, and a high negative correlation with neuroticism [5]. It has also been considered a meta-behavioral variable [6], closely associated with self-regulated learning [7,8], but also with emotional self-regulation [9]. Recent research has shown the effect of the student’s level of self-regulation on achievement emotions, based on the Self- vs. External- Regulated Learning (SRL vs. ERL) Theory [10,11].

The interactive model of emotion regulation [12,13,14] assumes that high levels of affective well-being occur when there is successful emotion regulation. Individuals experience and continuously modulate their emotional responses. Therefore, this model focuses on volitional activities; emotion regulation consists of the behaviors that a person uses after being exposed to a personally significant event that provokes an emotional response [14]. In this model, emotion regulation is understood to be the control over types of emotions that we experience, when we experience them, and the degree to which we experience them [13]. Control over these emotions is influenced both by a person’s regulatory behaviors and by the context that triggers the emotional experience. For this reason, generating high levels of affective well-being and emotion regulation in response to an emotion-triggering event is easier in some situations and more difficult in others.

Successful emotion regulation is often reflected in higher levels of positive emotions than negative emotions [12]. This does not mean the complete elimination of negative emotions. The negative effect (as a subjective experience of negative emotions) is necessary and adaptive in certain situations. For example, fear motivates a person to participate in safer behaviors; frustration may be necessary when working toward difficult goals. Successful emotion regulation processes may also involve, paradoxically, the triggering of negative emotions [15], as well as concurrent positive and negative emotions [16]. Nonetheless, in general, reports of higher affective well-being indicate more successful emotion regulation [17].

Previous research has shown self-regulation behavior to be significantly associated with coping strategies [18], learning approaches [19,20], resilience [21] and with emotions themselves [22]; it can have a regulating or dysregulating value [23,24]. Self-regulation behavior has also demonstrated its mediating value in mindfulness treatment with university students [25]. However, the effect of self-regulation behavior on positive vs. negative emotions, in accordance with the degree of stress elicited by a situation, has not been sufficiently established. This is the aspect analyzed in the present study. 

### 1.2. Level of Stress in Academic Situations, as a Contextual Variable 

The transactional model of stress [26,27,28] focuses on the subject’s interaction with the environment, on how he or she faces the demands that are continuously imposed by this environment. This interaction takes on meaning according to the subject’s (basically social) assessment of the environmental demands. The subject assesses events using cognitions that are the object of unintentional learning—such as in language acquisition, or the internalization of cultural patterns. The model can be summarized using the following schema:Environmental demands in terms of their implications. If this external, situational demand is unimportant, it does not affect the subject, and therefore does not have physical, personal or social implications. It does not trigger an emotional response. If it is important, however, it becomes a stress situation for the subject, prompting him or her to focus attention on it, and begin a process of assessment and response preparation for adaptation. Academic situations in the classroom, study time or testing can then be considered under this schema, classifiable according to the level of personal effort that each situation requires. Objectively, there are situations that trigger stress with such frequency, intensity and duration that they demand a more effortful response than others. However, the stress-eliciting potential also depends on mediating personal variables involved in processing these stimuli [29].Processes of assessing the stressful situation. The primary assessment is the subject’s own evaluation of the situation itself, what is involved in terms of risk, threat to his or her well-being or even survival. This is closely connected to the secondary assessment. In this assessment the subject evaluates the available resources, whether personal/social, or economic/institutional, for coping with the environmental demand. According to this assessment, the subject will consider whether he or she has resources and the capacity to meet the demand. The interplay of these two assessments determine whether the situation is considered: (1) a challenge: The subject considers that his or her resources are sufficient for handling the situation. The situation mobilizes the subject’s resources and generates feelings of efficacy and achievement; or (2) a threat: The subject considers that his or her resources are insufficient to handle the situation, and feels overwhelmed by it. Clear anxiety responses appear, and the subject’s coping activity is affected. He or she becomes ineffective, with reactions ranging from chaotic, frenetic, unorganized activity, to simple inactivity. The defining element is the perceived impossibility of control, essential to explanation of this pathology.

The subject’s level of self-regulation when interacting with the situation must be taken into account, given that this variable predicts others like resilience [21], type of coping strategies used and a final state of engagement-burnout [30]. The coping process can be approached from two directions, in terms of what the subject seeks to remedy, either his/her own perception and feelings, or the situation itself. Thus emerges two distinct types of coping, ideally complementary to each other: (1) Problem-focused coping. Oriented toward modifying the situation and changing it into something that does not suppose a threat to the subject. The process leads to a reinforcement of the subject’s role in his or her environment, both socially and personally, and encourages an outlook of self-efficacy in the face of future challenges. (2) Emotion-focused coping. In this case the subject does not modify the situation, considering that it is not possible at the present time, or perhaps at any time. This type of coping is present in pathologies where the subject avoids facing the problem situation, and seeks to distance himself/herself, or uses a nonreality tactic like denial or transferring to someone else their own responsibility in what happens. In short, it is a nonfunctional approach to managing the demands placed on the subject by real life.

Level of stress in a university academic situation refers to the degree of perceived stressfulness of a stress-triggering situation, according to students. In order to evaluate the degree or level of academic stress to which students are exposed, some researchers have made taxonomies of psychological situations [31,32,33]. Although there are fewer taxonomies that graduate the level of stress elicited by academic stressors (in comparison to health stressors), at least one study has established different types of academic stressors or situations [34]. In this classic study, the final exam was established as the situation with the greatest level of stress for students, followed by the partial exam. Excessive assignment workload was also a medium-high source of stress. Class attendance offered the lowest level of stress, as long as the teacher did not ask questions orally in class.

Other more recent taxonomies have established stress triggers that are inherent in the way the teaching and learning process is developed [35,36,37]. Organizational and functional aspects of teaching that directly affect students and produce stress symptoms, such as classroom climate, how homework is managed and organized, the importance of the content, or the assessment system itself, can be considered stress factors from the teaching process. On the other hand, stress factors from the learning process come from the student’s way of self-organizing and learning, his/her available social support, and personal anxiety level.

### 1.3. Positive vs. Negative Achievement Emotions as a Correlate of Emotional Well-Being in University Students

Unlike psychological well-being, which is more general and has been subject to much analysis in previous research [38,39,40], the study of emotional well-being has focused on emotions as a substantive element of subjective well-being. In the university setting, previous research has shown that a higher level of emotional well-being in university students was associated with a greater experience of positive emotions [41,42,43], while stress was associated with less psychological well-being and greater negative emotionality [44,45]. 

According to the control-value theory of achievement emotions [46,47], emotionality (positive vs. negative) proves to be an essential element in emotional well-being, being associated with other factors [48,49,50,51,52]. Recent research has contributed evidence regarding positive and negative emotionality in association with testing situations [53]. It has also recognized the effect of boredom on commitment and well-being [54]. From a broader perspective, positive and negative emotions have been shown to predict different coping strategies and, ultimately, the university student’s state of engagement vs. burnout. This contribution can in turn help clarify the functional mechanisms of perfectionism, in both its adaptive or maladaptive forms [30].

### 1.4. Aims and Hypotheses

The aim of this research was to analyze the joint effect of university students’ levels of self-regulation (1–low, 2–medium and 3–high), and their level of perceived stress in three academic situations (1—class, 2—study time and 3—testing), on the type of emotionality they experience (positive or negative). The following hypotheses were established: (1) according to the data from previous evidence, an increase in level of student self-regulation would predispose higher levels of positive emotionality (enjoyment, hope and pride) and lower levels of negative emotionality (boredom, anger, anxiety, shame and hopelessness); however, a decrease in level of self-regulation would produce the opposite. (2) In complementary fashion, an increase in the level of situational stress from the stress-eliciting academic situation would be accompanied by higher levels of negative emotionality and lower levels of positive emotionality; a decrease in the stress level of the situation would produce the opposite. This evidence is of great interest in understanding the extent to which student and contextual characteristics are able to predispose a certain type and degree of achievement emotions in university students.

## 2. Materials and Methods

### 2.1. Participants

A total of 520 students from two Spanish public universities voluntarily participated in this research study. The sample was composed of students majoring in psychology and primary education; 81.7% were women and 18.3% were men. The age range was 19–25, and mean age was 23.38 years (*sd* = 0.72). Sampling was incidental, non-probabilistic, because only nine of the professors who were invited from the different universities of the project agreed to participate with their students in this research, during one academic year (2018–2019). The students voluntarily completed self-reports, in the context of each academic subject (or specific teaching-learning process).

### 2.2. Instruments

*Self-regulation*. We used a short, validated version of the self-regulation questionnaire [55] for university students [56]. Through 17 items on a five-point answer scale, it provides information about the individual’s self-regulation behaviors. Reliability and validity values are adequate [57]. Sample items from the scale factors are: goal setting (I usually keep track of my progress toward my goals); perseverance (I am able to resist temptation); decision making (when it comes to deciding about a change, I feel overwhelmed by the choice) and learning from mistakes (I usually only have to make a mistake one time in order to learn from it). Internal consistency was acceptable for the total questionnaire (α = 0.86) and for the factors of goal setting-planning (α = 0.79), decision making (α = 0.72), perseverance (α = 0.73) and learning from mistakes (α = 0.72). 

Level of academic stress perceived in different situations. The students completed a Likert-type self-report, on an answer scale of (1 = low; 2 = medium; 3 = high), for the degree of threat in each situation: (1) class, (2) study and (3) testing. The situations to be rated appeared in random order. Internal consistency was acceptable for the total scale (α = 0.81) The three contexts of class, study time and testing have different elements that elicit stress; degree of stressfulness can reasonably be measured using the three classic behavioral parameters of frequency, intensity and duration. These parameters have a multiplicative effect toward determining the probable level of perceived stress triggered by an academic situation [34]. See Table 1:Frequency of stress triggered by the event. This refers to how often an event exposes the student to stress triggers that require adaptive effort. Any new situation involves an adaptive effort. Prior research has delimited the factors of situational academic stress [58], but has not specified the stress-eliciting situations. It can be assumed that the more often one is exposed to a novel stimulus—which involves effort and adaptive adjustment—the greater the potential to elicit stress.Intensity of stress triggered by the event. This refers to an event’s with force to evoke emotional responses, or the strength of the reactions it can elicit, whether positive or negative. We assume that the higher the level of emotional intensity triggered by the event, the greater its potential for eliciting a certain response.Duration of stress triggered by the event. This is measured by the objective amount of time that the person is exposed to the stimulus eliciting a response. A greater amount of exposure time is conceptualized as a longer duration of the response-eliciting stimulus.

*Achievement emotions*. Achievement emotions were assessed using a validated Spanish version [59,60,61] of the achievement emotions questionnaire [62], with adequate reliability and construct validity values [11]. This questionnaire was developed within the context of a research program (quantitative and qualitative) that analyzes student’s emotions experienced in academic achievement situations [63]. The instrument measures several discrete emotions that arise in the three main situations of academic performance: in class, study time and doing tests and exams. Three sections correspond to the three situations of class, study and testing, respectively. The class-related emotions scale (CRE) uses 80 items to measure class-related enjoyment, hope, pride, anger, anxiety, shame, hopelessness and boredom. The learning-related emotions scale (LRE) measures the same eight emotions in study situations, using 75 items. The test emotions scale (TE) measures these emotions in testing situations, using 77 items. The three scales each contain three subscales for measuring respectively the emotions that occur before, during, or after the academic situation being assessed. The scales measure the individual student’s typical emotional reactions to achievement situations, in other words, trait achievement emotions. If desired, it is possible to measure the emotions experienced in a particular academic subject (course-specific emotions), or in specific situations at specific times (state achievement emotions), by adapting the instructions to the AEQ. Some sample items are: emotions at the start of study (I have an optimistic view toward studying); during study time (because I’m bored I get tired sitting at my desk) and when finishing study (I am so happy about the progress I made that I am motivated to continue studying). The internal consistency of the class scale is good (Alpha = 0.904; Part 1 = 0.803, Part 2 = 0.853). Internal consistency of the total study scale is adequate (Alpha = 0.939; Part 1 = 0.880, Part 2 = 0.864). Internal consistency of the testing scale is correct (Alpha = 0.913; Part 1 = 0.870, Part 2 = 0.864). This self-report instrument assesses both type (positive vs. negative) and intensity (from 1 = none to 5 = very strong) of emotions experienced.

### 2.3. Procedure

This investigation falls within the framework of a broader research project and was approved by the ethics committees of the universities involved (Ref. 2018.170). The students voluntarily completed the questionnaires, after signing their informed consent. Validated questionnaires were used, in an online format, through the e-Coping with Stress platform [64]. At the start of the school year (2018–2019), students completed the self-regulation questionnaire. At the end of the school year, they completed the three scales on achievement emotions, in April (class situation), in May (study situation) and June (testing situation). The same students were asked to evaluate the degree of stress that each situation triggered (class, study and testing) in order to hypothesize a graduated level of stress in each situation. 

### 2.4. Data Analysis

Using an ex post facto design, we performed descriptive analyses, Pearson bivariate correlations, prediction analyses using structural equation models (SEM) and analyses of variance (ANOVAs and 3 × 3 MANOVAs), using IBM-SPSS v. 25 (New York, NY, USA). First, the database was checked for any incomplete cases; these were eliminated from the data. We did not conduct bivariate correlational analyses, due to their limitations in establishing multiple linear prediction. We performed multivariate analysis, testing data assumptions for the statistical methods used (multivariate normal distribution and between-group homogeneity of variance). Structural validity analysis was performed using IBM-AMOS statistical program, v. 23.0 (New York, NY, USA) [65] for Windows; this program was also used to construct the structural prediction model, specifically, verification of the structural linear prediction hypothesis (path analysis) [66]; Bentler, 1995). Comparative fit index (CFI) and Root Mean Square Error of Approximation (RMSEA) were used to interpret the Confirmatory Factor Analysis (CFA) and the fit of the structural equation model (SEM). Acceptable and close fit to the data were identified through the Normed Fit Index (NFI), the Goodness of Fit Index (GFI), the Comparative Fit Index (CFI), Incremental Fit Index (IFI), and Relative Fit Index (RFI) and Tuker Lewin Index (TLI) values equal to or more than 0.90 and 0.95, respectively [67]; and RMSEA values equal to or below 0.08 and 0.05, respectively [68]. To qualify direct effects, we used the beta coefficient cutoffs established in the research: less than 0.05; too small to be meaningful; above 0.05, small but meaningful; above 0.10, moderate and above 0.25, large [69]. We also used Kenny’s definition [70] of an indirect effect as the product of two effects. Accordingly, we proposed the following: an educationally meaningful, small indirect effect = 0.003, moderate indirect effect= 0.01 and large = 0.06.

## 3. Results

### 3.1. Description and Linear Associations Between Types of Achievement Emotions

A significant bivariate association appeared between SR and positive emotions (*r* = 0.431; *p* < 0.001) and between SR and negative emotions (*r* = −0. 374, *p* < 0.001). In the case of situational stress, it was also significantly associated with positive emotions (*r* = −0.119; *p* < 0.001) and with negative emotions (*r* = 0.290; *p* < 0.001). The descriptive results showed mean values of achievement emotions where positive emotions stand out above the negative emotions (positive emotions = 3.42 (0.68); negative emotions = 2.25 (0.66); *t*(1313) = 36,829, *p* < 0.001). Complementarily, positive associations were significant for emotions of both types, as were negative associations, with the opposite sign. See Table 2.

### 3.2. Structural Prediction of Level of SR and Level of Stress for Achievement Emotions

Predictive linear analyses revealed a significant path analysis model or SEM, where the level of self-regulation had greater predictive weight over positive and negative emotions than did the level of situational stress (result (default model): chi-square = 2173.155; degrees of freedom (65–32): 33; *p* < 0.001; NFI = 0.906; RFI = 0.902; IFI = 0.903; TLI = 0.912; CFI = 0.908; RMSEA = 0.08; Hoelter = 0.202 (*p* < 0.05) and 218 (*p* < 0.01)). In this model, the level of student regulation (GRUPSR) and the level of situational stress (SITUATION) have been considered observable independent variables. The type of achievement emotions (positive vs. negative) was considered a latent dependent variable. This structural prediction model has been used because it permits the analysis of direct and indirect effects between the variables, even if there is no intermediate mediating variable in the model.

The direct linear effects revealed direct prediction relationships where level of self-regulation had greater potential and significance for positively determining level of positive emotions (0.45) and negatively determining level of negative emotions (−0.45). Level of situational stress also had an effect, though less erful, for predicting positive emotions (−0.17) and negative emotions (0.19). The indirect linear effects gave evidence of the positive effect of level of self-regulation on positive emotions and its negative effect on negative emotions, while the inverse effect appeared for situations. See Figure 1 and Table 3 and Table 4.

### 3.3. Effects of Self-Regulation Level and Situational Stress Level on Emotionality

Different effects on the types of positive and negative emotions were seen in the analyses of variance (simple and multiple), where personal self-regulation level (3 levels) × situational stress level (3 levels) were taken as independent variables.

In this case, level of self-regulation (SR) had a general, significant main effect on achievement emotions (*F*(4,2468) = 81,881 (Pillai), *p* < 0.001; *r^p^* = 0.117; power = 1.0). Level of situational stress also showed this effect (SIT; *F*(4,2468) = 46,967 (Pillai), *p* < 0.001; *r^p^* = 0.070; power = 1.0). The partial effects also gave similar results. Specifically, at a higher level of self-regulation (SR), higher significant levels of positive emotionality were verified (*F*(2,1234) = 146,128 (Pillai), *p* < 0.001; *r^p^* = 0.191; power = 1.0; 3 (high SR) > 2 (medium SR) > 1 (low SR)) and lower levels of negative emotionality (*F*(2,1234) = 107,139 (Pillai), *p* < 0.001; *r^p^* = 0.148; pow = 1.0; 3 (high SR) < 2 (medium SR) < 1 (low SR)). Regarding situations of stress, there was also a significant partial effect that determined the level of negative emotions (*F*(2,1234) = 62,758 (Pillai), *p* < 0.001; *r^p^* = 0.092; pow = 1.0; 3 (test) > 2 (study) > 1 (class)) and positive emotions (*F*(2,1234) = 42,752 (Pillai), *p* < 0.001; *r^p^* = 0.062; pow = 1.0; 3 (test) < 2 (study) < 1 (class)).

Regarding the effect on specific achievement emotions, there was a general, significant, main effect of level of self-regulation (SR) on achievement emotions (*F*(16,2456) = 22,219 (Pillai), *p* < 0.001; *r^p^* = 0.126; pow = 1.0). The partial effects gave evidence of the significant effect of *SR* level on each of the emotions, with greater power for positive emotions like enjoyment (*F*(2,1234) = 92,657, *p* < 0.001; *r^p^* = 0.131; pow = 1.0; 3 > 2 > 1), hope (*F*(2,1234) = 170,763, *p* < 0.001; *r^p^* = 0.217; power = 1.0; 3 > 2 > 1) and pride (*F*(2,1234) = 102,033, *p* < 0.001; *r^p^* = 0.142; pow = 1.0; 3 > 2 > 1). There was also a significant inverse relation in negative emotions, such as boredom/relief (*F*(2,1234) = 32,901, *p* < 0.001; *r^p^* = 0.051; pow = 1.0; 1 > 2 > 3), anger (*F*(2,1234) = 66,197, *p* < 0.001; *r^p^* = 0.051; pow = 1.0; 1 > 2 > 3), anxiety (*F*(2,1234) = 76,694, *p* < 0.001; *r^p^* = 0.111; pow = 1.0; 1 > 2 > 3), shame (*F*(2,1234) = 76,674, *p* < 0.001; *r^p^* = 0.105; pow = 1.0; 1 > 2 > 3) and hopelessness (*F*(2,1234) = 98,098, *p* < 0.001; *r^p^* = 0.105; pow = 1.0; 1 > 2 > 3).

Level of situational stress (SIT) also showed a good deal of explanatory power in producing emotions (*F*(16,2456) = 72,447 (Pillai), *p* < 0.001; *r^p^* = 0.070; pow = 1.0). The partial effects showed a significant effect of level of situational stress on each of the emotions, with greater power for negative emotions, such as boredom/relief (*F*(2,1234) = 342,081, *p* < 0.001; *r^p^* = 0.357; power = 1.0; 3 > 2 > 1), anger (*F*(2,1234) = 23,274, *p* < 0.001; *r^p^* = 0.036; power = 1.0; 3 > 2 > 1), anxiety (*F*(2,1234) = 113,468, *p * < 0.001; *r^p^* = 0.113; pow = 1.0; 3 > 2 > 1), shame (*F*(2,1234) = 17,625, *p* < 0.001; *r^p^* = 0.068; pow = 1.0; 3 > 2 > 1) and hopelessness (*F*(2,1234) = 10.413, *p* < 0.001; *r^p^* = 0.017; power = 1.0; 3 > 2 > 1). For positive emotions, effects with less statistical power also appeared, in emotions like enjoyment (*F*(2,1234) = 36,358, *p* < 0.001; *r^p^* = 0.056; power = 1.0; 2 > 1 > 3), hope (*F*(2,1234) = 25,118, *p* < 0.001; *r^p^* = 0.039; power = 1.0; 2 > 1 > 3) and pride (*F*(2,1234) = 47,761, *p* < 0.001; *r^p^* = 0.122; power = 1.0; 3 > 2 > 1). Consequently, the highest level of enjoyment occurred in the study situation, while shame occurred more often in the classroom situation. However, deactivating emotions (boredom, relief) were the emotions most often found across all three levels of self-regulation. No significant interaction effects appeared. See Table 5 and Figure 2.

## 4. Discussion

The results of this study confirmed the first hypothesis, referring to a significant effect of the level of self-regulation for increasing levels of positive emotions and reducing the level of negative emotions. These results support the role of self-regulation as a meta-behavioral variable that affects emotion regulation during learning [9,71,72,73], in addition to regulating behavior itself [74]. These results are consistent with much prior evidence that has shown how a higher level of regulation is an antecedent to a greater number of positive emotions and a smaller number of negative emotions, while the opposite occurs with a low level of regulation [11,72]. 

The second hypothesis has also been confirmed, in that a higher level of perceived situational stress was interdependent with the level of negative emotionality produced by the situation [75]. It has been shown that a higher level of perceived situational stress is accompanied by an increase in negative emotionality and a decrease in positive emotionality, with the subsequent loss of emotional well-being [39,76]. In fact, students are likely to perceive a situation as eliciting higher/lower stress based on the correlates of negative/positive emotions that they experience; while the degree of greater/lesser emotional well-being would be associated with a greater/lesser experience of positive emotionality. The previous evidence has shown a linear relationship between the level of perceived stress—dependent on regulatory factors during the teaching and learning process—and the negative vs. positive emotionality experienced [11]. 

Other specific aspects relating to particular emotions are worthy of analysis. For example, while positive emotions (enjoyment, hope and pride) were produced in the study situation, negative emotions (anger, anxiety and hopelessness) occurred mainly in the testing situation [77]. The emotion of shame appeared most strongly in the class situation, probably because of the need to participate and speak in public [78]. These differences are of interest because they give us information about the specific emotional profile of students, allowing us to establish intervention strategies and psychoeducational support for university teachers and students [79].

The deactivating emotions (boredom and relief) were used in all three levels of self-regulation and across the three situations, appearing to indicate that they show a similar behavioral pattern, less dependent on the specific situation. Previous research has indicated that boredom—as a negative deactivating emotion—has its own behavior and is a precursor to disengagement and burnout [30,80,81]. Negative emotions have also appeared as precursors to risk behaviors, substance abuse and mental health problems [45]. 

For all these reasons, the results also allow us to specify the predictions of Pekrun’s theory [82,83,84,85,86,87], establishing how variability in achievement emotions is delimited by personal factors and the academic situational context.

### Limitations and Future Research

The present research study has several limitations. First, the sample does not have a cross cultural component, since only students from Spanish universities have participated. The lack of data with university students from different countries and cultures does not allow for the necessary generalization of results. This aspect should be addressed in future studies, to demonstrate an inter-cultural invariance of achievement emotions [88]. In addition, the study does not provide information on possible combinations between student characteristics and the three specific situations of university learning. This aspect has already been documented in other studies, where the combinations between the student’s self-regulation characteristics and teaching context characteristics consistently predispose certain results [11,18]. On the other hand, the value of mediating variables such as personality factors and maladaptive perfectionism also needs to be incorporated into the analysis of these relationships [89]. It is also important for future research to evaluate the emotional reactivity of students, as a mediating factor in the topographic profile of these emotions [90,91] and in emotional dysregulation [92].

## 5. Conclusions 

The present research led to two clear conclusions. Once again, it confirmed the importance of knowing university students’ pre-existing level of behavioral self-regulation, as a presage variable [93], because it indicates the type of emotions students were most likely to have during the learning process or different academic situations [93].

On the other hand, these results show that there was a progressive ladder of stress perceived by university students, with different types of specific emotionality attached. The testing situation is stressful because it involves a greater amount of negative emotionality, like anxiety or despair, and lesser amounts of positive emotionality, thereby affecting the level of self-regulated learning [94,95]. The study situation, with its medium level of stress, allows certain positive emotions like enjoyment [96] and confidence [97]. The class situation involves certain positive emotions, as well as certain negative emotions like shame, especially when there are tasks for which one does not feel competent, like speaking in public or working in a team [78].

## 6. Implications

All these results have clear implications for psychoeducational guidance at university, in the line of preventing academic stress [98,99,100]. It is essential that teachers become aware of these results, in order to properly balance their use of the different learning situations [101]. For example, excessive use of testing situations—with its heavy load of negative emotionality—can hinder emotions of enjoyment and academic behavioral confidence during the learning process [102]. Similarly, a class design that involves stressful activities does not encourage satisfaction or enjoyment in learning. Consequently, educational guidance professionals ought to pay attention not only to cognitive and meta-cognitive processes, but also to students who lack a high level of SR in learning. Attention to student diversity includes thinking about university students who learn with a medium or low level of self-regulation, if we do not want academic avoidance and dropout to continue to proliferate in our university systems [103,104]. 

## Figures and Tables

**Figure 1 ijerph-17-04293-f001:**
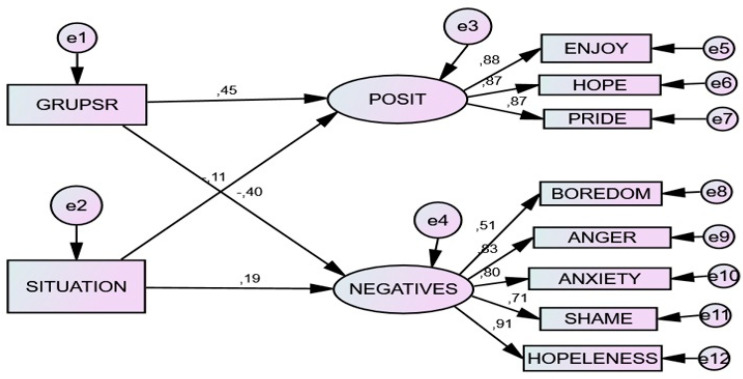
Predictive structural relationship of self-regulation level and situational stress level on achievement emotions. Note: GRUPSR = self-regulation Level: 1 (low), 2 (medium) and 3 (high). SITUATION = 1 (class), 2 (study) and 3 (test). POSIT = positive achievement emotions (enjoy, hope and pride) and NEGAT = negative achievement emotions (boredom, anger, anxiety, shame and hopelessness).

**Figure 2 ijerph-17-04293-f002:**
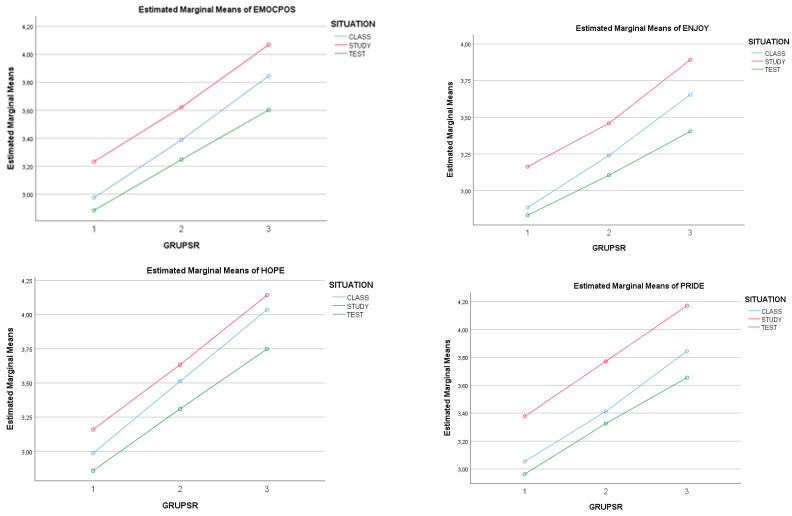
Effects of the SR level and situational stress level for specific achievement emotions. Note: GRPSR= group of level in self-regulation (1 = low, 2 = medium and 3 = high) and SITUATION = class, study and test.

**Table 1 ijerph-17-04293-t001:** Level of threat perceived in academic situations.

Situation	Parameters of Stress	Heading	Degree of Threat: Theoretical	Hypothesized	Degree of Threat: Empirical	Khon and Fracer
	F(x)	I(x)	D(x) = Rank	Range (1–3)	Mean and SD	Range (25–100)
1. Class	3 x	1 x	2 = 6	1 low	1.57 (.60)	39
2. Study	3 x	2 x	2 = 12	2. medium	2.37 (.64)	82
3. Testing	3 x	3 x	2 = 18	3. high	2.71 (.45)	100

Note. F = frequency of threat; I = intensity of threat; D = duration of threat and 3 = high; 2 = medium and 1 = low.

**Table 2 ijerph-17-04293-t002:** Correlation between achievement emotions (*n* = 520).

	Enjoyment	Hope	Pride	Boredom	Anger	Anxiety	Shame	Mean (sd)
Enjoyment								3.27 (0.70)
Hope	0.768 **							3.47 (0.76)
Pride	0.799 **	0.758 **						3.49 (0.75)
Boredom/Relief	−0.323 **	−0.325 **	−0.214 **					2.72 (0.05)
Anger	−0.333 **	−0.400 **	−0.265 **	0.544 **				2.01 (0.73)
Anxiety	−0.250 **	−0.388 **	−0.195 **	0.510 **	0.622 **			2.58 (0.82)
Shame	−0.186 *	−0.343 **	−0.227 **	0.139 *	0.531 **	0.598 **		2.08 (0.84)
Hopelessness	−0.324 **	−0.504 **	−0.374 **	0.380 **	0.766 **	0.706 **	0.695 **	1.91 (0.79)

Note. ** *p* < 0.001; * *p* < 0.01.

**Table 3 ijerph-17-04293-t003:** Standardized linear direct effects.

	SR Level	Situation Level	Positive	Negative
Positive	0.453	−0.109		
Negative	−0.402	0.186		
Enjoyment			0.882	
Hope			0.874	
Pride			0.870	
Boredom/Relief				0.506
Anger				0.830
Anxiety				0.802
Shame				0.710
Hopelessness				0.907

**Table 4 ijerph-17-04293-t004:** Standardized linear indirect effects.

	SR Level	Situation Level	Positive	Negative
Positive				
Negative				
Enjoyment	0.400	−0.096		
Hope	0.396	−0.095		
Pride	0.394	−0.095		
Boredom/Relief	−0.365	0.094		
Anger	−0.334	0.154		
Anxiety	−0.323	0.149		
Shame	−0.285	0.132		
Hopelessness	−0.365	0.168		

**Table 5 ijerph-17-04293-t005:** Level of academic emotions according to the Self-Regulation level and Situation.

SR	Low (*n* = 335)				Medium (*n* = 598)				High (*n* = 310)				
**Situation**	**Class**	**Study**	**Testing**	**Mean**	**Class**	**Study**	**Testing**	**Mean**	**Class**	**Study**	**Testing**	**Mean**	**Total**
	*n* = 104	*n* = 215	*n* = 99	*n* = 418	*n* = 114	*n* = 188	*n* = 108	*n* = 410	*n* = 117	*n* = 195	*n* = 103	*n* = 415	*n* = 1243
Emotions													
Positive	2.97 (0.59)	3.38 (0.54)	3.84 (0.61)	3.39 (0.64)	3.23 (0.65)	3.62 (0.54)	4.06 (0.68)	3.63 (0.66)	2.88 (0.68)	3.24 (0.60)	3.60 (0.62)	3.23 (0.68)	3.41 (0.68)
Negative	2.43 (0.63)	2.05 (0.56)	1.69 (0.54)	2.06 (0.63)	2.59 (0.68	2.16 (0.59)	1.81 (0.83)	2.19 (0.49)	2.78 (0.56)	2.51 (0.55)	2.27 (0.56)	2.26 (0.66)	2.26 (0.66)
Positive													
Enjoyment	2.88 (0.63)	3.16 (0.61)	2.83 (0.63)	2.99 (0.66)	3.23 (0.62)	3.45 (0.57)	3.10 (0.67)	3.26 (0.63)	3.65 (0.72)	3.89 (0.62)	3.40 (0.67)	3.65 (0.70)	3.27 (0.70)
Hope	2.98 (0.66)	3.15 (0.72)	2.85 (0.77)	3.00 (0.73)	3.51 (0.56)	3.63 (0.64)	3.31 (0.66)	3.48 (0.63)	4.03 (0.63)	4.14 (0.67)	3.74 (0.73)	3.97 (0.70)	3.47 (0.76)
Pride	3.05 (0.70)	3.37 (0.77)	2.96 (0.75)	3.13 (0.76)	3.41 (0.72)	3.77 (0.62)	3.32 (0.74)	3.42 (0.65)	3.84 (0.68)	4.17 (0.64)	3.65 (0.67)	3.89 (0.70)	3.49 (0.71)
Negative													
Boredom/Relief	2.75 (0.87)	2.67 (0.85)	3.48 (0.82)	2.97 (0.91)	2.27 (0.79)	2.25 (0.80)	3.67 (0.74)	2.72 (1.0)	1.90 (0.80)	1.79 (0.76)	3.64 (0.86)	2.44 (1.1)	2.72 (1.0)
Anger	2.19 (0.73)	2.27 (0.79)	2.50 (0.68)	2.32 (0.74)	1.86 (0.66)	1.94 (0.71)	2.18 (0.67)	1.99 (0.69)	1.57 (0.62)	1.61 (0.67)	1.91 (0.66)	1.69 (0.66)	2.01 (0.73)
Anxiety	2.51 (0.72)	2.98 (0.68)	3.30 (0.81)	2.94 (0.81)	2.18 (0.65)	2.60 (0.61)	2.91 (0.81)	2.55 (0.76)	1.78 (0.62)	2.28 (0.69)	2.62 (0.87)	2.23 (0.80)	2.58 (0.82)
Shame	2.57 (0.91)	2.64 (0.82)	2.22 (0.89)	2.47 (0.89)	2.19 (0.80)	2.09 (0.73)	1.83 (0.72)	2.04 (0.87)	1.79 (0.75)	1.81 (0.77)	1.60 (0.70)	1.74 (0.75)	2.08 (0.84)
Hopelessness	2.14 (0.74)	2.43 (0.87)	2.40 (0.84)	2.33 (0.83)	1.75 (0.60)	1.91 (0.70)	1.98 (0.70)	1.87 (0.70)	1.40 (0.55)	1.55 (0.71)	1.59 (0.79)	1.52 (0.70)	1.91 (0.79)

Note. SR level = levels of self-regulation (1 = low, 2 = medium and 3 = high) and SITUATION = levels of situational stress (1 = class; 2 = study and 3 = test).

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
