# Peer review of "Effect of Levels of Self-Regulation and Situational Stress on Achievement Emotions in Undergraduate Students: Class, Study and Testing"

_ijerph, 2020, doi:10.3390/ijerph17124293_

Round 1

Reviewer 1 Report

In general, I feel the study’s topic is important and the study itself has the potential to be published, however, more efforts are still needed.

  1. The language should be checked with the professional proofreader. There is even non-English expression in the paper, for example “ex post-facto” is showed twice in the paper.
  2. The “Aim and Hypotheses” is simply a copied excerpt from Abstract, which is totally unacceptable. The author shall rephrase either Abstract or Aim and Hypotheses.
  3. The authors claim, in their own words “(1) … self-regulation would predispose … positive emotionality and … negative emotionality, and vice versa; (2) … situational stress would be accompanied by … negative emotionality and … positive emotionality, and vice versa. ”. However, they only have cross-sectional data, so it is not possible to test the “vice versa”. You at least need to run a cross-lagged model with two wave data in order to examine the reciprocal effect. Moreover, in their regression model, they only checked uni-directional path. Surely, the correlations are bi-directionally, but this can not support your arguments.
  4. The introduction in “Level of stress in learning situations, as a contextual variable” is not sufficient. Please open up more after “Some researchers have made taxonomies of psychological situations (Edwards, & Templeton, 2005; Funder, 2006, 2016).”
  5. The table 1 is not consistent with what they say. Please check the carefully: Why there is “=” after 2? Above F, I, the descriptions are not following what they said. There are many other problems, please check them.
  6. The Cronbach alpha is not provided.
  7. You didn’t check the mediation effect, why do you claim you found “indirect effect”?
  8. The results you reported in the texts are not matched with the results you reported in the table. Please check them carefully.
  9. “Predictive analyses” is a bard wording, please rephrase.
  10. The Figure 1 contains non-English abbreviations, and they were not noted.
  11. The results in “Effects of Self-Regulation Level and Situational Stress Level on emotionality” are easy to be confused. What do 1, 2, 3 refer to in “3>2>1” and many others? It seems they are not consistent.
  12. The Figure 2 has a few problems. First, it is not in the correct order: enjoy comes before than EMOCNEG. Second, no plots for Stress and Emotions. Third, there are too many plots, I have to say, it is better to put them into the Supplementary Documents.

Author Response

In general, I feel the study’s topic is important and the study itself has the potential to be published, however, more efforts are still needed.

0) The language should be checked with the professional proofreader. There is even non-English expression in the paper, for example “ex post-facto” is showed twice in the paper.

Answer: The full text has been reviewed by a foreign translator.  The expression ex post facto is a Latin expression. It has been italicized, as it is taken directly from Latin. It is a type of design that is thus described in the psychological literature of research designs, as it is a type of design in which there is no experimental manipulation, but the data is manipulated by selection, after the occurrence of the event to be studied (Ato , López and Benavente, 2013)

1)The “Aim and Hypotheses” is simply a copied excerpt from Abstract, which is totally unacceptable. The author shall rephrase either Abstract or Aim and Hypotheses.

Answer:  The Objective and Hypothesis section has been revised and reformulated

2) The authors claim, in their own words “(1) … self-regulation would predispose … positive emotionality and … negative emotionality, and vice versa; (2) … situational stress would be accompanied by … negative emotionality and … positive emotionality, and vice versa. ”. However, they only have cross-sectional data, so it is not possible to test the “vice versa”. You at least need to run a cross-lagged model with two wave data in order to examine the reciprocal effect. Moreover, in their regression model, they only checked uni-directional path. Surely, the correlations are bi-directionally, but this can not support your arguments.

Answer: Text has been adjusted for greater precision; This statement refers to the levels of interdependence that ANOVAS and MANOVAS provide

3) The introduction in “Level of stress in learning situations, as a contextual variable” is not sufficient. Please open up more after “Some researchers have made taxonomies of psychological situations (Edwards, & Templeton, 2005; Funder, 2006, 2016).”

Answer: This section has been completed. Thank you.

4) The table 1 is not consistent with what they say. Please check the carefully: Why there is “=” after 2? Above F, I, the descriptions are not following what they said. There are many other problems, please check them.

Answer:   Thank you. It has been revised for clarity.

5) The Cronbach alpha is not provided.

Answer: It has been inserted

6) You didn’t check the mediation effect, why do you claim you found “indirect effect”?

Answer: Structural predictive analysis using AMOS allows to check the linear effect of indirect mediation. The expression has been adjusted.

7) The results you reported in the texts are not matched with the results you reported in the table. Please check them carefully.

Answer: Thank you. Errors have been reviewed and corrected. The meaning of levels 1,2,3 has also been inserted in both variables.

8) “Predictive analyses” is a bard wording, please rephrase.

Answer:   It has been reformulated: predictive linear structural análisis

9) The Figure 1 contains non-English abbreviations, and they were not noted.

Answer: I have corrected

10) The results in “Effects of Self-Regulation Level and Situational Stress Level on emotionality” are easy to be confused. What do 1, 2, 3 refer to in “3>2>1” and many others? It seems they are not consistent.

Answer:   Information has been inserted

11) The Figure 2 has a few problems. First, it is not in the correct order: enjoy comes before than EMOCNEG. Second, no plots for Stress and Emotions. Third, there are too many plots, I have to say, it is better to put them into the Supplementary Documents.

Answer: It has been rearranged and adjusted to a horizontal format. The information is relevant to present it in figures

Reviewer 2 Report

The research article "Effect of Levels of Self-Regulation and Situational Stress on Achievement Emotions in Undergraduate Students: Class, Study and Testing" aims to investigate associations between stress, emotions, and self-regulation among university students. Concerning the research topic the research generally fits to the aims and scope of the journal. However, the paper has several weaknesses and I cannot recommend a publication in its current form. The author(s) might get the opportunity to revise their study and submit again after handling the problems described below.

Introduction

What is meant by "This research study, therefore, is situated at the molar level of psychoeducational analysis" ? (line 32/33) --- please elaborate.

Important psychological concepts (e.g., emotions, stress, self-regulation behavior) have not been defined and described sufficiently.

The concept of "emotional well-being" is named in the introduction section, however it is not operationalized in the method section.

All in all, the introduction section lacks a well founded theoretical modeling of the assumed relationships between variables of interest (emotions, stress, self-regulation behavior; moderation effects vs. mediation effects). The authors just report some existing empirical findings on assumed relationships, however, a well founded theoretical model is completely missing (see, for example, Richard S. Lazarus: Stress and Emotion: A New Synthesis).

Aims and hypotheses

It is not clear why the author(s) differentiate only between three levels of Self-Regulation (1-low, 2-medium, 3-high) and do not use a continuous variable.

Further, it is not clear why the author(s) differentiate between the mentioned three academic situations (1-class, 2- study time and 3-testing).

The deduction of the hypotheses is not well founded in the theoretical part of the paper (see above).

Method, Participants

How was the sample recruited? Why was incidental/non-probabilistic sampling chosen?

Please move "Procedure" section after "Participants" section and before "Instruments" section.

Method, Instruments

Please provide sample items for the used instruments.

Please provide reliability measures for the used instruments.

Why only the "degree of stress" was assessed? The transactional model of stress and coping differentiates between stress perception (in terms of cognitive registration of a potential stressor) and stress appraisel (in terms of evaluation of the potential stressor as irrelevant, challenging or threatening) --- the subjective appraisal is essential on how strong the symptomatical stress outcome is.

It remains unclear how the stress measures (Frequency of stress triggered by the event, Intensity of stress triggered by the event, Duration of stress triggered by the event) were measured in detail and how the overall stress score was formed in detail.

It remains unclear what achievement emotions were assessed in detail.

Data analysis

Please describe which software (description and version) was used in detail.

Please provide information on what statistical methods are used in detail and what strategy of analyses was used to test the hypotheses.

Assumptions concerning endogeny or endogeneity of variables should be discussed.

A mathematical model / formula is missing; please provide equations.

Please provide information on data assumptions concerning the used statistical methods (e.g., multivariate normal distribution etc.).

Results

There are very high correlations between some emotion variables (e.g., Enjoyment & Hope, Enjoyment & Pride). Thus, the assumed scale structure should be verified first via confirmatory factor analysis before performing SEM analyses.

Figure 1 is not printed in good quality. Further, it is not clear what the abbreviations (GRUPSR, SITUACION, POSIT, NEGATIVES) mean.

Table 3: What means "SR LEVEL"?

Please provide the information from section "Effects of Self-Regulation Level and Situational Stress Level on emotionality" in a Table.

References on Table 5 are missing in the text. Please provide more information! If moderating effects are assessed, you should provide corresponding hypotheses and state clear what statistical method you use in order to assess moderating effects. If you assume moderating effects, you should discuss them theoretically and differentiate them fom possible mediating effects.

The variables in Tables and Figures should be named in real names, not in abbreviations.

Discussion, Limitations and future research

Why do you see the missing "cross cultural component" as a limitation? It is unclear why the inter-cultural invariance is important for further studies.

Concerning "English language and style": I am not a native speaker, but i think the manuscript needs a professional language proof.

Author Response

Introduction

  1. What is meant by "This research study, therefore, is situated at the molar level of psychoeducational analysis" ? (line 32/33) --- please elaborate.

Answer: It has been explained in the text.

2. Important psychological concepts (e.g., emotions, stress, self-regulation behavior) have not been defined and described sufficiently.

The concept of "emotional well-being" is named in the introduction section, however it is not operationalized in the method section.

3. All in all, the introduction section lacks a well founded theoretical modeling of the assumed relationships between variables of interest (emotions, stress, self-regulation behavior; moderation effects vs. mediation effects). The authors just report some existing empirical findings on assumed relationships, however, a well founded theoretical model is completely missing (see, for example, Richard S. Lazarus: Stress and Emotion: A New Synthesis).

Answer: These points have been reviewed and expanded. There are theoretical models that support research.

Aims and hypotheses

  1. It is not clear why the author(s) differentiate only between three levels of Self-Regulation (1-low, 2-medium, 3-high) and do not use a continuous variable.

Answer There are two reasons for choosing 3 SR variable level (low-medium-high):

1) is consistent with the levels proposed in the SRL vs ERL theory (de la fuente, 2017)

2) allows to perform SIMPLE AND MULTIPLE INFERENTIAL ANALYSIS

  1. Further, it is not clear why the author(s) differentiate between the mentioned three academic situations (1-class, 2- study time and 3-testing).

 Answer: From the definition of situational stress and the exposed classifications, it is assumed that there is a graduation of the levels of situational stress, from lowest to highest, in these three common situations in the university environment: class, study, exam

  1. The deduction of the hypotheses is not well founded in the theoretical part of the paper (see above).

Answer: Thanks. The foundation has been substantially improved.

Method

  1. How was the sample recruited? Why was incidental/non-probabilistic sampling chosen?

Answer: Participation has been expanded.

  1. Please move "Procedure" section after "Participants" section and before "Instruments" section.

Answer: In the field of Psychology, the usual order is this.

  1. Please provide sample items for the used instruments.

 Answer: Sample information has been expanded

  1. Please provide reliability measures for the used instruments.

Answer: Instrument reliability information has been inserted.

11.Why only the "degree of stress" was assessed? The transactional model of stress and coping differentiates between stress perception (in terms of cognitive registration of a potential stressor) and stress appraisel (in terms of evaluation of the potential stressor as irrelevant, challenging or threatening) --- the subjective appraisal is essential on how strong the symptomatical stress outcome is.

Answer: The text has been reformulated - more explicitly - in the theoretical framework of the Lazarus & Folkman (1984) transactional stress model.

12. It remains unclear how the stress measures (Frequency of stress triggered by the event, Intensity of stress triggered by the event, Duration of stress triggered by the event) were measured in detail and how the overall stress score was formed in detail.

Answer: The parameters of frequency, intensity and duration of the effort in the 3 situations to calculate the resulting threat level is a proper heuristic that has been contrasted with the empirical assessment of the students and the levels established by the Khon & Fracer (1986) model. 

13. It remains unclear what achievement emotions were assessed in detail.

 Answer: More information on the structure of the questionnaires has been inserted

Data analysis

14.Please describe which software (description and version) was used in detail.

Answer: This information has been inserted

  1. Please provide information on what statistical methods are used in detail and what strategy of analyses was used to test the hypotheses.

Answer: This information has been inserted

16.Assumptions concerning endogeny or endogeneity of variables should be discussed.

Answer: Thank you. This information had been taken for granted. Has been inserted

17.A mathematical model / formula is missing; please provide equations.

Answer: This information is in line 296 to 298

Please provide information on data assumptions concerning the used statistical methods (e.g., multivariate normal distribution etc.).

Answer: This informationa has been inserted

Results

18. They are very high correlations between some emotion variables (e.g., Enjoyment & Hope, Enjoyment & Pride). Thus, the assumed scale structure should be verified first via confirmatory factor analysis before performing SEM analyses.

Answer: The scales were previously tested using SEM analysis. The results have already been published and have been presented in the instruments section. Please, see:

https://www.mdpi.com/1660-4601/17/6/2106

19.Figure 1 is not printed in good quality. Further, it is not clear what the abbreviations (GRUPSR, SITUACION, POSIT, NEGATIVES) mean.

 Answer: The errors and the figure have been revised. A NOTE has been placed at the bottom of the figure.

20.Table 3: What means "SR LEVEL"?

Answer: An explanatory note has been inserted at the foot of the Table

20. Please provide the information from section "Effects of Self-Regulation Level and Situational Stress Level on emotionality" in a Table.

Answer: An explanatory opening paragraph has been inserted, for clarification.

21. References on Table 5 are missing in the text. Please provide more information! If moderating effects are assessed, you should provide corresponding hypotheses and state clear what statistical method you use in order to assess moderating effects. If you assume moderating effects, you should discuss them theoretically and differentiate them fom possible mediating effects.

Answer: In this analysis, there are no mediating or modulating effects. There are direct effects of VI -> VDs. Is consistent with the proposed inferential hypotheses

22. The variables in Tables and Figures should be named in real names, not in abbreviations.

Answer: It is possible to abbreviate if specified with a footnote of the Graph or Figure

                Discussion

23. Why do you see the missing "cross cultural component" as a limitation? It is unclear why the inter-cultural invariance is important for further studies.

Answer: This explication has been inserted in the test

  1. Concerning "English language and style": I am not a native speaker, but i think the manuscript needs a professional language proof.

Answer: The full text has been reviewed by an American translator, specialized in Educational Psychology.

Round 2

Reviewer 1 Report

Thanks for the revision, this version has been much improved.

However, a few points are still needed to be considered.

  1. I am still not convinced about your responses and the expression of "indirect effect". You said "Structural predictive analysis using AMOS allows to check the linear effect of indirect mediation. ". However, which are the mediators? Your figure and tables did not have any clue of it. What I have seen is the outcomes became discrete emotions instead of aggregated positive or negative emotion.
  2. At the very beginning, you stated your approach can assess person X situation relationships. However, you didn't find those interactions effects in your results. On the one hand, you didn't discuss this finding. Why there is no interaction in your findings? On the other hand, if this is an important question, why this has not been in your hypotheses?
  3. I have a little concern about your treating of the "SITUATION". In the study, this is actually a categorical variable (class, study, and test). However, you treated it as a continuous variable in your analysis. How will the results change if you treated it as a categorical variable?

Author Response

1)I am still not convinced about your responses and the expression of "indirect effect". You said "Structural predictive analysis using AMOS allows to check the linear effect of indirect mediation. ". However, which are the mediators? Your figure and tables did not have any clue of it. What I have seen is the outcomes became discrete emotions instead of aggregated positive or negative emotion.

1) Response: Thanks for the question. The linear prediction analysis allows us to verify the direct and indirect effects between variables. The prediction structural equation can be made up of latent variables or inferred (oval) or discrete observable (rectangles) constructs. For example, the model reveals a direct negative prediction effect of the variable the level of self-regulation (GRUPSR-oval) in the construct negative emotions (NEGATIVES-oval), but also an indirect prediction effect on discrete emotions (boredom,….-rectangles). That is, the indirect prediction effect calculated by AMOS can not only be performed between latent variables (ovals), with a latent mediating variable, but also considering the construct variable (oval) as the mediating variable and the observable variables (rectangles) as the predicted variables.

2)At the very beginning, you stated your approach can assess person x situation relationships. However, you didn't find those interactions effects in your results. On the one hand, you didn't discuss this finding. Why there is no interaction in your findings? On the other hand, if this is an important question, why this has not been in your hypotheses?

2) Response. The consideration of the person x situation analysis refers to the joint analysis of the weight that both variables have to predict, the dependent variable, in this case positive and negative emotions. It is not assumed that an interaction effect should appear (graphs in the form of X), but a main effect of each variable, although independently and not interactive (this is what happens). If we had assumed this hypothesis we would be assuming that there would be an interactive effect, that is to say that students with high levels of SR would have more positive emotionality in the classes and less negative emotionality in the exams; while the low ones in SR would have less positive emotions in the classes and more in the exams. The graphics would be cross-shaped (in X). This directionality is not real and has not been found in the previous evidence either. Yes, the independent effect of each of the variables. Therefore, an SR x Situation interaction hypothesis has not been proposed.

3) I have a little concern about your treating of the "SITUATION". In the study, this is actually a categorical variable (class, study, and test). However, you treated it as a continuous variable in your analysis. How will the results change if you treated it as a categorical variable?

3) Response: Analyzes have shown (1) the effects as a continuous variable (LINEAR PREDICTION ANALYSIS) and (2) subsequently as a categorical variable (ANOVAL AND MANOVAS INFERENTIAL ANALYSIS). Both analyzes have been performed to show, precisely, the not only predictive but also causal inferential effect of each level (3 x 3) of both variables (SR and SITUATION), on academic emotions (positive and negative).

I hope I have adequately clarified your doubts. Thank you.